# Binary matrix factorization on special purpose hardware

Osman Asif Malik[1]*, Hayato Ushijima-Mwesigwa[2], Arnab Roy[2], Avradip Mandal[2], Indradeep Ghosh[2]

**1** Department of Applied Mathematics, University of Colorado Boulder, Boulder, CO, United States of America, **2** Fujitsu Research of America, Inc., Sunnyvale, CA, United States of America

* osman.malik@colorado.edu

**Data Availability Statement:** All relevant data are within the paper and its Supporting information files.

**Funding:** The authors HUM, AR, AM and IG are employees of Fujitsu Research of America, Inc.,

## Abstract

Many fundamental problems in data mining can be reduced to one or more NP-hard combinatorial optimization problems. Recent advances in novel technologies such as quantum and quantum-inspired hardware promise a substantial speedup for solving these problems compared to when using general purpose computers but often require the problem to be modeled in a special form, such as an Ising or quadratic unconstrained binary optimization (QUBO) model, in order to take advantage of these devices. In this work, we focus on the important binary matrix factorization (BMF) problem which has many applications in data mining. We propose two QUBO formulations for BMF. We show how clustering constraints can easily be incorporated into these formulations. The special purpose hardware we consider is limited in the number of variables it can handle which presents a challenge when factorizing large matrices. We propose a sampling based approach to overcome this challenge, allowing us to factorize large rectangular matrices. In addition to these methods, we also propose a simple baseline algorithm which outperforms our more sophisticated methods in a few situations. We run experiments on the Fujitsu Digital Annealer, a quantum-inspired complementary metal-oxide-semiconductor (CMOS) annealer, on both synthetic and real data, including gene expression data. These experiments show that our approach is able to produce more accurate BMFs than competing methods.

## Introduction

Many fundamental problems in data mining consist of discrete decision making and are combinatorial in nature. Examples include feature selection, data categorization, class assignment, identification of outlier instances, $k$-means clustering, combinatorial extensions of support vector machines, and consistent biclustering, to mention a few [1]. In many cases, these underlying problems are NP-hard, and approaches to solving them therefore dependent on heuristics. Recently, researchers have been exploring different computing paradigms to tackle these NP-hard problems, including quantum computing and the development of dedicated special purpose hardware. The Ising and quadratic unconstrained binary optimization (QUBO)

which is the developer of the Fujitsu Digital Annealer used in the experiments. OAM worked as an intern at Fujitsu Research of America, Inc., when the research in this paper was carried out. The funder provided support in the form of salaries for authors OAM, HUM, AR, AM and IG, but did not have any additional role in the study design, data collection and analysis, decision to publish, or preparation of the manuscript. The specific roles of these authors are articulated in the 'author contributions' section.

**Competing interests:** The authors HUM, AR, AM and IG are employees of Fujitsu Research of America, Inc., which is the developer of the Fujitsu Digital Annealer used in the experiments. OAM worked as an intern at Fujitsu Research of America, Inc., when the research in this paper was carried out. The authors helped file a patent application on behalf of Fujitsu Research of America, Inc., which is based on the work in the present paper. This does not alter our adherence to PLOS ONE policies on sharing data and materials.

models are now becoming unifying frameworks for the development of these novel types of hardware for combinatorial optimization problems.

Binary matrix factorization is an NP-hard combinatorial problem that many computational tasks originating from a wide range of applications can be reformulated into. These applications include areas such as data clustering [2–6], pattern discovery [7, 8], dictionary learning [9], collaborative filtering [10], association rule mining [11], dimensionality reduction [12], and image rendering [13]. As such, any advances in solving the binary matrix factorization problem, can potentially lead to breakthroughs in various application domains.

In this paper we show how the aforementioned hardware technologies, via the QUBO framework, can be used for binary matrix factorization. As Moore's law comes to an end [14], investigating how such post-Moore's law technologies can be used is an important task. This is especially true for primitives like binary matrix factorization which are used in data mining tasks that continue to grow ever larger and more complex. We make the following contributions in this paper:

- Provide two QUBO formulations for one variant of the binary matrix factorization problem. To the best of our knowledge, these are the first methods specifically designed for solving binary matrix factorization on quantum and quantum-inspired hardware to appear in the literature. We additionally propose a simple baseline method which outperforms our more sophisticated methods in a few situations.

- Show how constraints that are useful in clustering tasks can easily be incorporated into the QUBO formulations.

- Present a sampling heuristic for factorizing large rectangular matrices.

- Conduct experiments on both synthetic and real data on the Fujitsu Digital Annealer. These experiments suggest that our method is able to achieve higher accuracy than competing methods in the kind of binary matrix factorization we consider.

## Binary matrix factorization

Let $A \in \{0, 1\}^{m \times n}$ be a matrix with binary entries. For a positive integer $r \leq \min(m, n)$, the rank-$r$ binary matrix factorization (BMF) problem is

$$\min_{U, V} \|A - UV^\top\|_F^2 \quad \text{s.t.} \quad U \in \{0, 1\}^{m \times r}, \ V \in \{0, 1\}^{n \times r}. \tag{1}$$

We discuss other definitions of BMF that appear in the literature in the section Related work.

**Example 1** (Exact BMF). Define matrices

$$A \stackrel{\text{def}}{=} \begin{bmatrix} 1 & 1 & 0 \\ 1 & 1 & 1 \\ 0 & 0 & 1 \end{bmatrix}, \quad U \stackrel{\text{def}}{=} \begin{bmatrix} 0 & 1 \\ 1 & 1 \\ 1 & 0 \end{bmatrix}, \quad V \stackrel{\text{def}}{=} \begin{bmatrix} 0 & 1 \\ 0 & 1 \\ 1 & 0 \end{bmatrix}.$$

Then $A = UV^\top$ is an exact BMF of $A$.

## The QUBO framework

Let $Q \in \mathbb{R}^{n \times n}$ be a matrix. A QUBO problem takes the form

$$\min_x x^\top Q x \quad \text{s.t.} \quad x \in \{0, 1\}^n. \tag{2}$$

The tutorial by Glover et al. [15] is a good introduction to this problem which also discusses some of its many applications.

## The Digital Annealer

The Fujitsu Digital Annealer (DA) is a hardware accelerator for solving fully connected QUBO problems. Internally the hardware runs a modified version of the Metropolis–Hastings algorithm [16, 17] for simulated annealing. The hardware utilizes massive parallelization and a novel sampling technique. The novel sampling technique speeds up the traditional Markov Chain Monte Carlo (MCMC) method by almost always moving to a new state instead of being stuck in a local minimum. As explained in [18], in the DA, each Monte Carlo step takes the same amount of time, regardless of accepting a flip or not. In addition, when accepting the flip, the computational complexity of updating the effective fields is constant regardless of the connectivity of the graph. The DA also supports Parallel Tempering (replica exchange MCMC sampling) [19] which improves dynamic properties of the Monte Carlo method. In our experiments, we use the DA coupled with software techniques as our main QUBO solver.

## Notation

Bold upper case letters (e.g. $A$) denote matrices, bold lower case letters (e.g. $x$) denote vectors, and lower case regular and Greek letters (e.g. $x, \lambda$) denote scalars. Subscripts are used to indicate entries in matrices and vectors. For example, $a_{ij}$ is the entry on position $(i, j)$ in $A$. A $*$ in a subscript is used to denote all entries along a dimension. For example, $a_{i*}$ and $a_{*j}$ are the $i$th row and $j$th column of $A$, respectively. We use $1$, $0$ and $I$ to denote a matrix of ones, a matrix of zeros, and the identity matrix, respectively. Subscripts are also used to indicate the size of these matrices. For example, $1_{m \times n}$ is an $m \times n$ matrix of all ones, $1_n \stackrel{\text{def}}{=} 1_{n \times n}$ is an $n \times n$ matrix of all ones, and $I_n$ is the $n \times n$ identity matrix. These subscripts are omitted when the size is obvious. Superscripts in parentheses will be used to number matrices and vector (e.g. $A^{(1)}, A^{(2)}$). The matrix Kronecker product is denoted by $\otimes$. The function vec($\cdot$) takes a matrix and turns it into a vector by stacking all its columns into one long column vector. The function diag($\cdot$) takes a vector input and returns a diagonal matrix with that vector along the diagonal. Semicolon is used as in Matlab to denote vertical concatenation of vectors. For example, if $u \in \mathbb{R}^m$ and $v \in \mathbb{R}^n$ are column vectors, then $[u; v]$ is a column vector of length $m + n$. The Frobenius norm of a matrix $A$ is defined as

$$\|A\|_{\text{F}} \stackrel{\text{def}}{=} \sqrt{\sum_{ij} a_{ij}^2}.$$

For positive integers $n$, we use the notation $[n] \stackrel{\text{def}}{=} \{1, 2, \ldots, n\}$.

## Related work

The most popular methods for BMF are the two by Zhang et al. [2, 3]. Their first approach alternates between updating $U$ and $V$ until some convergence criteria is met. It incorporates a penalty which encourages the entries of $U$ and $V$ to be near 0 or 1. At the end of the algorithm, the entries of $U$ and $V$ are rounded to ensure they are exactly 0 or 1. Their second approach initializes $U$ and $V$ using nonnegative matrix factorization. For each factor matrix, a threshold is then identified, and values in the matrix below and above the threshold are rounded to 0 and 1, respectively.

Koyutürk and Grama [11] develop a framework called PROXIMUS, which decomposes binary matrices by recursively using rank-1 approximations, which results in a hierarchical representation. Shen et al. [7] provide a linear program formulation for the rank-1 BMF problem and provide approximation guarantees. Ramírez [9] presents methods for BMF applied to binary dictionary learning. Kumar et al. [13] provide faster approximation algorithms for BMF as well as a variant of BMF for which inner products are computed over the finite field of two elements (GF(2)). Diop et al. [20] propose a variant of BMF for binary matrices which takes the form $A \approx \Phi(UV^{\top})$, where $U$ and $V$ are binary, and $\Phi$ is a nonlinear sigmoid function. They use a variant of the penalty approach by [2, 3] to compute the decomposition.

Boolean matrix decomposition, which is also referred to as binary matrix decomposition by some authors, is similar to the BMF in (1), but an element $(UV^{\top})_{ij}$ is computed via

$$(UV^{\top})_{ij} = \bigvee_{k=1}^{r} u_{ik} v_{jk}$$

instead of the standard inner product, where $\bigvee$ denotes disjuction. Some works that consider Boolean matrix decomposition include [4–6, 8, 10, 12, 21, 22]. For a theoretical comparison between BMF, Boolean matrix factorization and a variant of BMF computed over GF(2), we refer the reader to the recent paper by DeSantis et al. [23].

There are previous works that use special purpose hardware to solve linear algebra problems. O'Malley and Vesselinov [24] discuss how linear least squares can be solved via QUBO formulations on D-Wave quantum annealing machines. They consider both the case when the solution vector is restricted to being binary and when it is real valued. The real valued case is handled by representing entries in the solution vector using a fixed number of bits. O'Malley et al. [25] consider a nonnegative/binary factorization of a real valued matrix of the form $A \approx WH$, where $W$ has nonnegative entries and $H$ is binary. To compute this factorization, they use an alternating least squares approach by iteratively alternating between solving for $W$ and $H$. When solving for $H$, they use the QUBO formulation from [24] for the corresponding binary least squares problem and do the computation on a D-Wave quantum annealer. Drawing inspiration from [24], Ottaviani and Amendola [26] propose a QUBO formulation for low-rank nonnegative matrix factorization and also implement it on a D-Wave machine. They too use an alternating least squares approach combined with real number representations similar to those in [24]. Borle et al. [27] show how the Quantum Approximate Optimization Algorithm (QAOA) framework can be used for solving binary linear least squares. Their paper includes experiments run on an IBM Q machine. Unlike our paper, none of the works [24–27] consider binary matrix factorization. Additionally, an important difference between our work and the decomposition techniques developed in [25, 26] is that those papers update the factor matrices in an alternating fashion. Our two QUBO formulations, by contrast, solve for *both* factor matrices at the same time, which may help avoid the issue of getting stuck in local minima that alternating algorithms are susceptible to. However, we do incorporate alternating optimization as a post-processing step in our experiments since this can sometimes further improve the quality of the solutions that come from solving the QUBO formulations. See the sections Handling large rectangular matrices and Experiments for details.

There has also been a large body of research on utilizing special purpose hardware for data clustering problems, for example [28–32] to name a few. A recent paper by Şeker et al. [33] performs a comprehensive computational study comparing the DA to multiple state-of-the-art solvers on multiple different combinatorial optimization problems. They find that the DA performs favorably compared to the other solvers, particularly on large problem instances.

## QUBO formulations for BMF

Writing out the objective in (1), we get

$$\|\boldsymbol{A} - \boldsymbol{U}\boldsymbol{V}^\top\|_F^2 = \|\boldsymbol{A}\|_F^2 - 2\sum_{ijk} a_{ij} u_{ik} v_{jk} + \sum_{ijkk'} u_{ik} u_{ik'} v_{jk} v_{jk'}, \tag{3}$$

where the summations are over $i \in [m]$, $j \in [n]$, and $k, k' \in [r]$. Our goal is to reformulate this into the QUBO form in (2). The fourth order term in (3) stops us from directly writing (3) on the quadratic form in (2). We can get around this by introducing appropriate auxiliary variables and penalties.

### Formulation 1

For our first formulation, we introduce auxiliary variables

$$w_{ij}^{(k)} \stackrel{\text{def}}{=} u_{ik} v_{jk} \quad \text{for } i \in [m], \ j \in [n], \ k \in [r]. \tag{4}$$

We arrange these variables into $m \times n$ matrices $\boldsymbol{W}^{(k)} = (w_{ij}^{(k)})$. An equivalent formulation to (1) is then

$$\min_{\boldsymbol{U},\boldsymbol{V},\{\boldsymbol{W}^{(k)}\}} \|\boldsymbol{A}\|_F^2 - 2\sum_{ijk} a_{ij} w_{ij}^{(k)} + \sum_{ijkk'} w_{ij}^{(k)} w_{ij}^{(k')}$$

$$\text{s.t. } \boldsymbol{U} \in \{0,1\}^{m \times r}, \ \boldsymbol{V} \in \{0,1\}^{n \times r}, \ (4) \text{ satisfied.} \tag{5}$$

To incorporate the constraints (4) in the QUBO model, we express them as a penalty instead. A standard technique for this [15] is to use a penalty function $f : \{0,1\}^3 \to \mathbb{R}$ defined via

$$f(a, b, c) \stackrel{\text{def}}{=} bc - 2ba - 2ca + 3a. \tag{6}$$

Notice that $f(a, b, c) = 0$ if $a = bc$ and $f(a, b, c) \geq 1$ otherwise. Letting $\lambda$ be a positive constant, a penalty variant of (5) is

$$\min_{\boldsymbol{U},\boldsymbol{V},\{\boldsymbol{W}^{(k)}\}} \|\boldsymbol{A}\|_F^2 - 2\sum_{ijk} a_{ij} w_{ij}^{(k)} + \sum_{ijkk'} w_{ij}^{(k)} w_{ij}^{(k')} + \lambda \sum_{ijk} f(w_{ij}^{(k)}, u_{ik}, v_{jk})$$

$$\text{s.t. } \boldsymbol{U} \in \{0,1\}^{m \times r}, \ \boldsymbol{V} \in \{0,1\}^{n \times r}, \ \boldsymbol{W}^{(k)} \in \{0,1\}^{m \times n} \text{ for all } k \in [r]. \tag{7}$$

**Proposition 2.** *Suppose* $\lambda > 2r\|\boldsymbol{A}\|_F^2$. *A point* $(\boldsymbol{U}, \boldsymbol{V}, \{\boldsymbol{W}^{(k)}\})$ *minimizes* (5) *if and only if it minimizes* (7).

*Proof.* For brevity, we denote the objectives in (5) and (7) by $\text{OBJ}_1$ and $\text{OBJ}_2$, respectively. Setting all entries in the matrices $\boldsymbol{U}, \boldsymbol{V}, \boldsymbol{W}^{(1)}, \ldots, \boldsymbol{W}^{(r)}$ to zero would yield an objective value of $\text{OBJ}_2(\boldsymbol{U}, \boldsymbol{V}, \{\boldsymbol{W}^{(k)}\}) = \|\boldsymbol{A}\|_F^2$. Moreover,

$$-2\sum_{ijk} a_{ij} w_{ij}^{(k)} + \sum_{ijkk'} w_{ij}^{(k)} w_{ij}^{(k')} \geq -2r\|\boldsymbol{A}\|_F^2,$$

so any point $(\boldsymbol{U}, \boldsymbol{V}, \{\boldsymbol{W}^{(k)}\})$ that violates (4) would satisfy $\text{OBJ}_2(\boldsymbol{U}, \boldsymbol{V}, \{\boldsymbol{W}^{(k)}\}) > \|\boldsymbol{A}\|_F^2$ and therefore could not be a minimizer of (7). Any minimizer of (7) must therefore satisfy (4).

Suppose $p^* \stackrel{\text{def}}{=} (\boldsymbol{U}^*, \boldsymbol{V}^*, \{\boldsymbol{W}^{(k)*}\})$ minimizes (5). Since $p^*$ satisfies (4), all penalty terms in (7) are zero for this point, and therefore $\text{OBJ}_1(p^*) = \text{OBJ}_2(p^*)$. $p^*$ is also a minimizer of (7). If it was not, there would be a minimizer $p^\dagger$ of (7) for which $\text{OBJ}_2(p^\dagger) < \text{OBJ}_2(p^*)$ and which would

satisfy (4). $p^{\dagger}$ would therefore be a feasible solution for (5) and it would satisfy $\mathrm{OBJ}_1(p^{\dagger}) = \mathrm{OBJ}_2(p^{\dagger}) < \mathrm{OBJ}_1(p^*)$, which contradicts the optimality of $p^*$.

Suppose $p^{\dagger} \stackrel{\text{def}}{=} (\boldsymbol{U}^{\dagger}, \boldsymbol{V}^{\dagger}, \{\boldsymbol{W}^{(k)\dagger}\})$ minimizes (7). Then $p^{\dagger}$ satisfies (4) and is therefore a feasible solution to (5) satisfying $\mathrm{OBJ}_1(p^{\dagger}) = \mathrm{OBJ}_2(p^{\dagger})$. $p^{\dagger}$ is also a minimizer of (5). If it was not, there would be a minimizer $p^*$ of (5) which would satisfy $\mathrm{OBJ}_2(p^*) = \mathrm{OBJ}_1(p^*) < \mathrm{OBJ}_1(p^{\dagger}) = \mathrm{OBJ}_2(p^{\dagger})$, contradicting the optimality of $p^{\dagger}$.

Since (1) and (5) are equivalent, Proposition 2 implies that the matrices $\boldsymbol{U}$ and $\boldsymbol{V}$ we get from minimizing (7) are minimizers of (1) when $\lambda$ is sufficiently large.

We now state the QUBO formulation of (7). Define $\boldsymbol{u} \stackrel{\text{def}}{=} \mathrm{vec}(\boldsymbol{U})$, $\boldsymbol{v} \stackrel{\text{def}}{=} \mathrm{vec}(\boldsymbol{V})$, and $\boldsymbol{w}^{(k)} \stackrel{\text{def}}{=} \mathrm{vec}(\boldsymbol{W}^{(k)})$ for each $k \in [r]$, and let

$$\boldsymbol{x} \stackrel{\text{def}}{=} \begin{bmatrix} \boldsymbol{u} ; & \boldsymbol{v} ; & \boldsymbol{w}^{(1)} ; & \boldsymbol{w}^{(2)} ; & \cdots ; & \boldsymbol{w}^{(r)} \end{bmatrix}, \tag{8}$$

where $\boldsymbol{x}$ is a column vector of length $(m + n + mn)r$. Furthermore, define the QUBO matrix as

$$\boldsymbol{Q} \stackrel{\text{def}}{=} \begin{bmatrix} \boldsymbol{Q}^{(1)} & \boldsymbol{Q}^{(2)} \\ \boldsymbol{0} & \boldsymbol{Q}^{(3)} \end{bmatrix} \in \mathbb{R}^{(m+n+mn)r \times (m+n+mn)r}, \tag{9}$$

where

$$\boldsymbol{Q}^{(1)} \stackrel{\text{def}}{=} \frac{\lambda}{2} \begin{bmatrix} \boldsymbol{0}_{mr} & \boldsymbol{I}_r \otimes \boldsymbol{1}_{m \times n} \\ \boldsymbol{I}_r \otimes \boldsymbol{1}_{n \times m} & \boldsymbol{0}_{nr} \end{bmatrix},$$

$$\boldsymbol{Q}^{(2)} \stackrel{\text{def}}{=} -2\lambda \begin{bmatrix} \boldsymbol{I}_r \otimes \boldsymbol{1}_{1 \times n} \otimes \boldsymbol{I}_m \\ \boldsymbol{I}_{nr} \otimes \boldsymbol{1}_{1 \times m} \end{bmatrix},$$

$$\boldsymbol{Q}^{(3)} \stackrel{\text{def}}{=} \boldsymbol{1}_r \otimes \boldsymbol{I}_{mn} - 2 \; \mathrm{diag}\left((\boldsymbol{1}_{r \times 1} \otimes \boldsymbol{I}_{mn}) \; \mathrm{vec}(\boldsymbol{A})\right) + 3\lambda \boldsymbol{I}_{mnr}.$$

**Proposition 3**. *With $\boldsymbol{x}$ and $\boldsymbol{Q}$ defined as in (8) and (9), respectively, the problem (7) can be written as*

$$\min_{\boldsymbol{x}} \|\boldsymbol{A}\|_{\mathrm{F}}^2 + \boldsymbol{x}^{\top} \boldsymbol{Q} \boldsymbol{x} \quad s.t. \quad \boldsymbol{x} \in \{0, 1\}^{(m+n+mn)r}. \tag{10}$$

The proof is a straightforward but somewhat tedious calculation and is omitted. Although removing the constant $\|\boldsymbol{A}\|_{\mathrm{F}}^2$ does not affect the minimizer(s) of (10), it can serve as a useful target: If $\boldsymbol{x}^{\top} \boldsymbol{Q} \boldsymbol{x} = -\|\boldsymbol{A}\|_{\mathrm{F}}^2$, then we know that we have found a global minimum, provided that the condition on $\lambda$ in Proposition 2 is satisfied. Such a target value can be supplied to QUBO solvers like D-Wave's QBSolv to allow for early termination when the target is reached.

## Formulation 2

For our second formulation, we again consider (3) and introduce auxiliary variables

$$\tilde{u}_{i(kk')} \stackrel{\text{def}}{=} u_{ik} u_{ik'} \quad \text{for } i \in [m], \; k, k' \in [r],$$

$$\tilde{v}_{j(kk')} \stackrel{\text{def}}{=} v_{jk} v_{jk'} \quad \text{for } j \in [n], \; k, k' \in [r]. \tag{11}$$

We treat $\tilde{\boldsymbol{U}} = (\tilde{u}_{i(kk')})$ and $\tilde{\boldsymbol{V}} = (\tilde{v}_{j(kk')})$ as matrices of size $m \times r^2$ and $n \times r^2$, respectively, with $\tilde{\boldsymbol{u}}_{*(kk')}$ being the $(k + k'r)$th column of $\tilde{\boldsymbol{U}}$, with similar column ordering for $\tilde{\boldsymbol{V}}$. An

equivalent formulation to (1) is then

$$\min_{U,V,\tilde{U},\tilde{V}} \|A\|_{\mathrm{F}}^2 - 2\sum_{ijk} a_{ij} u_{ik} v_{jk} + \sum_{ijkk'} \tilde{u}_{i(kk')} \tilde{v}_{j(kk')} \tag{12}$$

$$\text{s.t. } U \in \{0,1\}^{m \times r}, \; V \in \{0,1\}^{n \times r}, \; (11) \text{ satisfied.}$$

We use the function $f$ defined in (6) to incorporate the constraints (11) in the objective. A penalty variant of (12) is

$$\min_{U,V,\tilde{U},\tilde{V}} \|A\|_{\mathrm{F}}^2 - 2\sum_{ijk} a_{ij} u_{ik} v_{jk} + \sum_{ijkk'} \tilde{u}_{i(kk')} \tilde{v}_{j(kk')}$$

$$+ \lambda \sum_{ikk'} f(\tilde{u}_{i(kk')}, u_{ik}, u_{ik'}) + \lambda \sum_{jkk'} f(\tilde{v}_{j(kk')}, v_{jk}, v_{jk'}) \tag{13}$$

$$\text{s.t. } U \in \{0,1\}^{m \times r}, \; V \in \{0,1\}^{n \times r}, \; \tilde{U} \in \{0,1\}^{m \times r^2}, \; \tilde{V} \in \{0,1\}^{n \times r^2}.$$

**Proposition 4**. *Suppose* $\lambda > 2r\|A\|_{\mathrm{F}}^2$. *A point* $(U, V, \tilde{U}, \tilde{V})$ *minimizes* (12) *if and only if it minimizes* (13).

The proof is similar to that for Proposition 2 and is omitted. Since (1) and (12) are equivalent, Proposition 4 implies that the matrices $U$ and $V$ we get from minimizing (13) are minimizers of (1) when $\lambda$ is sufficiently large.

We now state the QUBO formulation of (13). Define $u$ and $v$ as before. Furthermore, define $\tilde{u} \overset{\mathrm{def}}{=} \mathrm{vec}(\tilde{U})$, $\tilde{v} \overset{\mathrm{def}}{=} \mathrm{vec}(\tilde{V})$ and

$$y \overset{\mathrm{def}}{=} [\, u \, ; \quad v \, ; \quad \tilde{u} \, ; \quad \tilde{v} \,], \tag{14}$$

where $y$ is a column vector of length $(m+n)(r+r^2)$. Furthermore, define the QUBO matrix as

$$P \overset{\mathrm{def}}{=} \begin{bmatrix} P^{(1)} & P^{(2)} \\ 0 & P^{(3)} \end{bmatrix} \in \mathbb{R}^{(m+n)(r+r^2) \times (m+n)(r+r^2)}, \tag{15}$$

where

$$P^{(1)} \overset{\mathrm{def}}{=} \begin{bmatrix} \lambda \mathbf{1}_r \otimes I_m & -2I_r \otimes A \\ 0_{nr \times mr} & \lambda \mathbf{1}_r \otimes I_n \end{bmatrix},$$

$$P^{(2)} \overset{\mathrm{def}}{=} -2\lambda \begin{bmatrix} \mathbf{1}_{1 \times r} \otimes I_{mr} & 0_{mr \times nr^2} \\ 0_{nr \times mr^2} & \mathbf{1}_{1 \times r} \otimes I_{nr} \end{bmatrix} - 2\lambda \begin{bmatrix} I_r \otimes \mathbf{1}_{1 \times r} \otimes I_m & 0_{mr \times nr^2} \\ 0_{nr \times mr^2} & I_r \otimes \mathbf{1}_{1 \times r} \otimes I_n \end{bmatrix},$$

$$P^{(3)} \overset{\mathrm{def}}{=} \begin{bmatrix} 3\lambda I_{mr^2} & I_{r^2} \otimes \mathbf{1}_{m \times n} \\ 0_{nr^2 \times mr^2} & 3\lambda I_{nr^2} \end{bmatrix}.$$

**Proposition 5**. *With* $y$ *and* $P$ *defined as in* (14) *and* (15), *respectively, the problem* (13) *can be written as*

$$\min_y \|A\|_{\mathrm{F}}^2 + y^\top P y \quad s.t. \; y \in \{0,1\}^{(m+n)(r+r^2)}. \tag{16}$$

The proof is a straightforward and is omitted.

## Useful constraints for data analysis

In this section we show how certain constraints that are helpful for data mining tasks easily can be incorporated into the QUBO formulations. One approach to clustering of the rows and/or columns of a binary matrix $A$ is to compute a BMF $A \approx UV^\top$ and then use the information in $U$ and $V$ to build the clusters. This idea is used e.g. by [2, 3] for gene expression sample clustering and document clustering. For gene expression data, the rows of $A$ represent genes and the columns represent samples, e.g. from different people. An unsupervised data mining task on such a dataset could be to identify and cluster people based on if they have cancer or not. One way to do this is to compute a rank-2 BMF of $A$ and assign sample $j$ to cluster $k \in \{1, 2\}$ if $v_{jk} = 1$. In many cases, it is reasonable to require that each column belongs to precisely one cluster. For example, when clustering people based on if they have cancer or not, we want to assign every person to precisely one of two clusters. Such a requirement can be incorporated by enforcing that

$$\sum_k v_{jk} = 1 \quad \text{for all } j. \tag{17}$$

A penalty variant of this constraint is

$$\lambda \sum_j (1 - \sum_k v_{jk} + 2 \sum_{k<k'} v_{jk} v_{jk'}), \tag{18}$$

where $\lambda > 0$. Since $V$ is binary, the penalty is zero when (17) is satisfied, and at least $\lambda$ otherwise. This penalty can simply be added to the objectives in (7) and (13). As before, we can ensure that the penalized and constrained formulations have the same minimizers by choosing $\lambda > 2r\|A\|_F^2$.

The penalty (18) is straightforward to incorporate into either QUBO formulation. Define an $(m + n)r \times (m + n)r$ matrix

$$C \stackrel{\text{def}}{=} \begin{bmatrix} 0 & 0 \\ 0 & \lambda(1_r \otimes I_n - 2I_{nr}) \end{bmatrix}.$$

The QUBO formulations in (10) and (16) are easily modified to incorporate (18) by defining modified QUBO matrices

$$Q' \stackrel{\text{def}}{=} \begin{bmatrix} Q^{(1)} + C & Q^{(2)} \\ 0 & Q^{(3)} \end{bmatrix},$$

$$P' \stackrel{\text{def}}{=} \begin{bmatrix} P^{(1)} + C & P^{(2)} \\ 0 & P^{(3)} \end{bmatrix},$$

where the submatrices $Q^{(i)}$ and $P^{(i)}$ are defined as before.

## Handling large rectangular matrices

In this section, we present a strategy for handling large rectangular matrices. We consider the case when $A$ is $m \times n$ with $m \gg n$ and $n$ is of moderate size. These ideas also apply when $n \gg m$ and $m$ is of moderate size. Random sampling of rows and columns is a popular technique in numerical linear algebra for compressing large matrices. These compressed matrices are then

used instead of the full matrices in computations. For an introduction to this topic we recommend the survey by Mahoney [34].

A popular sampling approach is to sample according to the *leverage scores* of the matrix. Suppose $A \in \mathbb{R}^{m \times n}$ is a nonzero matrix, and let $B \in \mathbb{R}^{m \times \mathrm{rank}(A)}$ be an orthonormal matrix whose columns form a basis for range($A$). The leverage scores of $A$ are defined as $\ell_i(A) \stackrel{\text{def}}{=} \|B_{i*}\|_2^2$ for $i \in [m]$. $B$ can be computed via, e.g., the singular value decomposition (SVD). The cost $O(mn^2)$ of computing the SVD of $A$ is small compared to the cost of solving the BMF. If this cost proves to be too expensive, then there are techniques for estimating the leverage scores that only cost $O(mn \log m)$ [35]. When sampling rows of $A$ according to the leverage scores, we sample the $i$th row of $A$ with probability $p_i \stackrel{\text{def}}{=} \ell_i(A)/\mathrm{rank}(A)$ for $i \in [m]$. This definition guarantees that $\Sigma_i p_i = 1$. We use leverage score sampling as a heuristic for compressing $A$ by sampling $s \ll m$ of the rows of $A$ with replacement according to the distribution $(p_i)$ and putting these in a new matrix $A^{(s)} \in \mathbb{R}^{s \times n}$. We then compute a rank-$r$ BMF $A^{(s)} \approx U^{(s)}V^\top$. To get a BMF for the original matrix $A$, we then solve the binary least squares (BLS) problem

$$U \stackrel{\text{def}}{=} \arg \min_{U' \in \{0,1\}^{m \times r}} \|A - U'V^\top\|_{\mathrm{F}}^2, \tag{19}$$

where $V$ comes from the factorization of $A^{(s)}$. By expanding the objective, the problem in (19) can be written as $m$ independent BLS problems involving $r$ binary variables. These BLS problems can be solved via a QUBO formulation. As discussed in [24, 25], such a formulation is easy to derive by noting that the BLS objective can be written as

$$\|Mx - y\|_2^2 = x^\top(M^\top M - 2 \; \mathrm{diag}(y^\top M))x + \|y\|_2^2.$$

Setting $Q \stackrel{\text{def}}{=} (M^\top M - 2 \; \mathrm{diag}(y^\top M))$ gives us a QUBO objective as in (2). Alternatively, when $r$ is small, the optimal solution to each BLS problem can be found by simply testing all $2^r$ possible solutions. As an optional step after computing $U$, a few additional alternating BLS steps can be added. This is done by minimizing the objective in (19) in an alternating fashion, first solving for $V$ and treating $U$ as fixed, and then solving for $U$ and treating $V$ as fixed.

## Experiments

We found that Formulation 1 yields a lower decomposition error for a given number of iterations than Formulation 2 on the Fujitsu DA. We therefore use the former in our experiments and refer to it as "DA BMF" or just "DA" in the tables. Additionally, we try adding a few extra alternating BLS steps (as discussed in the section Handling large rectangular matrices) to the solutions we get from the DA. We do at most 20 alternating BLS solves, and whenever no improvement occurs after two consecutive BLS solves, we terminate. Since $r \leq 5$ in our experiments, we solve the BLS problems exactly by checking all possible solutions. We refer to this method as "DA+ALS BMF" or just "DA+ALS" in the tables. For some of the real datasets, we incorporate the constraint in the section Useful constraints for data analysis. For cases when $A$ is large and rectangular, we use the sampling technique in the section Handling large rectangular matrices. We will point out when the sampling and/or additional constraints are used.

We run our proposed method on the Fujitsu DA for a fixed number of 1e+9 iterations. Here, an iteration refers to one iteration of the for loop on line 5 in Algorithm 2 of [18]. We do not try to find an optimal number of iterations. We take this approach to avoid cherry picking a number of iterations that works great for each individual problem. By choosing a relative large number of iterations, we are also hoping to push the hardware to see how good solutions it can find.

As discussed by Glover et al. [15], although the penalty $\lambda$ needs to be sufficiently large to ensure that the constrained and penalized versions of our optimization problems have the same minimizers, setting $\lambda$ to a smaller value may improve the solution produced by a QUBO solver in practice. An intuitive explanation for this phenomenon is that a large $\lambda$ value gives a steeper optimization landscape which can make it difficult for a solver to escape local minima. We find this to be true when running our methods on the Fujitsu DA as well. We use $\lambda = 1$ in all our experiments since we found this to improve the solution quality, while at the same time avoiding constraint violations.

As mentioned in the section Related work, the two methods by Zhang et al. [2, 3] are the most popular for the variant of BMF we consider. We therefore use these methods for comparison in our experiments. We refer to them as "Penalized" and "Thresholded," respectively. For the penalized version, we use the Bmf method in the Nimfa Python library [36] available at http://nimfa.biolab.si. We leave all parameters to their default values, except the maximum number of iterations (max_iter) and the frequency of the convergence test (test_conv) which we both set to 1000 since we find that this improves performance substantially over the defaults in our experiments. We wrote our own implementation of the thresholded method since we could not find an existing implementation; see the section Implementation of thresholding method for BMF in S1 Text for details.

We also include a simple baseline method. The idea behind it is simple: If we seek a rank-$r$ BMF of $A$, we can find one by simply choosing the densest $r$ rows/columns in $A$. Alternatively, when $A$ has high density, we can approximate it by a rank-1 BMF with all entries equal to 1. This is clearly a very crude method, but it serves as a useful baseline and sanity check for the more sophisticated methods. See the section Details on baseline method in S1 Text for further details.

All experiment results are evaluated in terms of the following relative error measure: $\|A - UV^\top\|_F^2 / \|A\|_F^2$. The norm is squared since this is more natural for binary data: When $U$ and $V$ are binary, $\|A - UV^\top\|_F^2$ is the number of entries that are incorrect in the decomposition and $\|A\|_F^2$ is the number of nonzero entries in $A$.

## Synthetic data

For the first set of synthetic experiments, $A$ is generated in such a way that it has an exact rank-$r$ decomposition, for $r \in [5]$. Our algorithm for generating these matrices is described in the section Algorithm for generating binary matrices in S1 Text. We use the true rank as the target rank for the decompositions. Ideally, the different methods should therefore be able to find an exact decomposition. We generate $A$ to have $n = 30$ columns and $m \in \{30, 2000, 50000\}$ rows. When $m \geq 2000$, we use the sampling technique described in the section Handling large rectangular matrices for all our methods. We use a sample size of 30, so that $A^{(s)} \in \{0, 1\}^{30 \times 30}$. All experiments are repeated 10 times. Table 1 reports the mean relative error for these experiments.

For the second set of synthetic experiments, we draw each entry $a_{ij}$ independently from a Bernoulli distribution with probability of success $p \in \{0.2, 0.5, 0.8\}$. We generate $A$ with $n = 30$ columns and $m \in \{30, 2000, 50000\}$ rows, and use target ranks $r \in [5]$. When $m \geq 2000$, we use the sampling technique described in the section Handling large rectangular matrices for our methods with a sample size of $s = 30$. All experiments are repeated 10 times. Tables 2–4 report mean relative errors for each of the three different values of $p$.

In all synthetic experiments, the QUBO problem has $(30 + 30 + 30^2)r = 960r$ binary variables. This is also true for the large rectangular matrices due to the choice of sample size $s = 30$.

**Table 1. Mean relative error for synthetic _A_ with an exact decomposition.** The * symbol indicates methods we propose. Best results are underlined.

| Method | Target ranks _r_ (_m_ = 30) | | | | |
|---|---|---|---|---|---|
| | 1 | 2 | 3 | 4 | 5 |
| *DA | 0 | 0 | 0 | 0 | 0 |
| *DA+ALS | 0 | 0 | 0 | 0 | 0 |
| Penalized | 0 | 0 | 0 | 0.0170 | 0.0148 |
| Thresholded | 0 | 0 | 0 | 0.0052 | 0.0273 |
| *Baseline | 0.8265 | 0.8706 | 0.8157 | 0.7434 | 0.6977 |
| Method | Target ranks _r_ (_m_ = 2000) | | | | |
| | 1 | 2 | 3 | 4 | 5 |
| *DA | 0 | 0 | 0 | 0 | 0 |
| *DA+ALS | 0 | 0 | 0 | 0 | 0 |
| Penalized | 0 | 0 | 0 | 0 | 0.0361 |
| Thresholded | 0 | 0 | 0 | 0.0330 | 0.0594 |
| *Baseline | 0.9181 | 0.9075 | 0.8730 | 0.8101 | 0.7585 |
| Method | Target ranks _r_ (_m_ = 50000) | | | | |
| | 1 | 2 | 3 | 4 | 5 |
| *DA | 0 | 0 | 0 | 0 | 0 |
| *DA+ALS | 0 | 0 | 0 | 0 | 0 |
| Penalized | 0 | 0 | 0 | 0 | 0.0159 |
| Thresholded | 0 | 0 | 0.0240 | 0.0317 | 0.0632 |
| *Baseline | 0.8831 | 0.9088 | 0.8634 | 0.8127 | 0.7520 |

**Table 2. Mean relative error for synthetic _A_ for which $a_{ij} \sim$ Bernoulli(0.2).** The * symbol indicates methods we propose. Best results are underlined.

| Method | Target ranks _r_ (_m_ = 30) | | | | |
|---|---|---|---|---|---|
| | 1 | 2 | 3 | 4 | 5 |
| *DA | 0.9209 | 0.8565 | 0.8018 | 0.7643 | 0.7161 |
| DA-ALS | 0.9209 | 0.8565 | 0.8018 | 0.7637 | 0.7161 |
| Penalized | 0.9989 | 0.9230 | 0.8559 | 0.7875 | 0.7397 |
| Thresholded | 0.9555 | 0.8904 | 0.8409 | 0.7954 | 0.7609 |
| *Baseline | 0.9365 | 0.8787 | 0.8238 | 0.7734 | 0.7253 |
| Method | Target ranks _r_ (_m_ = 2000) | | | | |
| | 1 | 2 | 3 | 4 | 5 |
| *DA | 0.9902 | 0.9743 | 0.9659 | 0.9452 | 0.9484 |
| *DA+ALS | 0.9895 | 0.9727 | 0.9624 | 0.9403 | 0.9436 |
| Penalized | 1.0000 | 1.0000 | 0.9998 | 0.9918 | 0.9751 |
| Thresholded | 0.9990 | 0.9932 | 0.9777 | 0.9601 | 0.9368 |
| *Baseline | 0.9639 | 0.9281 | 0.8928 | 0.8577 | 0.8228 |
| Method | Target ranks _r_ (_m_ = 50000) | | | | |
| | 1 | 2 | 3 | 4 | 5 |
| *DA | 0.9914 | 0.9785 | 0.9624 | 0.9491 | 0.9421 |
| *DA+ALS | 0.9914 | 0.9785 | 0.9623 | 0.9489 | 0.9413 |
| Penalized | 1 | 1 | 1 | 0.9986 | 0.9839 |
| Thresholded | 0.9998 | 0.9951 | 0.9833 | 0.9661 | 0.9443 |
| *Baseline | 0.9660 | 0.9323 | 0.8985 | 0.8649 | 0.8313 |

**Table 3. Mean relative error for synthetic *A* for which $a_{ij} \sim$ Bernoulli(0.5).** The * symbol indicates methods we propose. Best results are <u>underlined</u>.

| Method | Target ranks *r* (*m* = 30) | | | | |
|---|---|---|---|---|---|
| | 1 | 2 | 3 | 4 | 5 |
| *DA | <u>0.7844</u> | <u>0.6773</u> | 0.6056 | 0.5536 | 0.5165 |
| *DA+ALS | <u>0.7844</u> | <u>0.6773</u> | <u>0.6054</u> | <u>0.5534</u> | <u>0.5146</u> |
| Penalized | 0.8606 | 0.7306 | 0.6611 | 0.6147 | 0.5749 |
| Thresholded | 0.7983 | 0.7170 | 0.6775 | 0.6875 | 0.6680 |
| *Baseline | 0.9519 | 0.9103 | 0.8679 | 0.8265 | 0.7870 |
| **Method** | **Target ranks *r* (*m* = 2000)** | | | | |
| | 1 | 2 | 3 | 4 | 5 |
| *DA | 0.8809 | 0.8234 | 0.7948 | 0.7540 | 0.7347 |
| *DA+ALS | 0.8628 | <u>0.8100</u> | <u>0.7888</u> | <u>0.7507</u> | <u>0.7301</u> |
| Penalized | 0.9802 | 0.9635 | 0.9371 | 0.8862 | 0.8478 |
| Thresholded | <u>0.8568</u> | 0.8346 | 0.8299 | 0.8311 | 0.8038 |
| *Baseline | 0.9651 | 0.9307 | 0.8964 | 0.8622 | 0.8282 |
| **Method** | **Target ranks *r* (*m* = 50000)** | | | | |
| | 1 | 2 | 3 | 4 | 5 |
| *DA | 0.8820 | 0.8214 | 0.7846 | 0.7550 | 0.7325 |
| *DA+ALS | 0.8634 | <u>0.8099</u> | <u>0.7807</u> | <u>0.7521</u> | <u>0.7324</u> |
| Penalized | 0.9912 | 0.9819 | 0.9565 | 0.9033 | 0.8770 |
| Thresholded | <u>0.8555</u> | 0.8358 | 0.8324 | 0.8418 | 0.8323 |
| *Baseline | 0.9664 | 0.9328 | 0.8992 | 0.8657 | 0.8322 |

**Table 4. Mean relative error for synthetic *A* for which $a_{ij} \sim$ Bernoulli(0.8).** The * symbol indicates methods we propose. Best results are <u>underlined</u>.

| Method | Target ranks *r* (*m* = 30) | | | | |
|---|---|---|---|---|---|
| | 1 | 2 | 3 | 4 | 5 |
| *DA | <u>0.2446</u> | 0.2288 | 0.2192 | 0.2097 | 0.2050 |
| *DA+ALS | <u>0.2446</u> | <u>0.2257</u> | <u>0.2129</u> | <u>0.2012</u> | <u>0.1916</u> |
| Penalized | 0.2446 | 0.2441 | 0.2539 | 0.2843 | 0.3346 |
| Thresholded | 0.2446 | 0.2904 | 0.4390 | 0.5078 | 0.5332 |
| *Baseline | 0.2446 | 0.2446 | 0.2446 | 0.2446 | 0.2446 |
| **Method** | **Target ranks *r* (*m* = 2000)** | | | | |
| | 1 | 2 | 3 | 4 | 5 |
| *DA | <u>0.2503</u> | 0.2550 | 0.2539 | 0.2590 | 0.2480 |
| *DA+ALS | <u>0.2503</u> | <u>0.2473</u> | <u>0.2459</u> | <u>0.2402</u> | <u>0.2384</u> |
| Penalized | 0.2503 | 0.2500 | 0.3202 | 0.4757 | 0.5743 |
| Thresholded | 0.2503 | 0.2625 | 0.6319 | 0.7702 | 0.7581 |
| *Baseline | <u>0.2503</u> | 0.2503 | 0.2503 | 0.2503 | 0.2503 |
| **Method** | **Target ranks *r* (*m* = 50000)** | | | | |
| | 1 | 2 | 3 | 4 | 5 |
| *DA | <u>0.2502</u> | 0.2546 | 0.2542 | 0.2632 | 0.2438 |
| *DA+ALS | <u>0.2502</u> | <u>0.2471</u> | <u>0.2447</u> | <u>0.2428</u> | <u>0.2363</u> |
| Penalized | 0.2502 | 0.2574 | 0.3220 | 0.4996 | 0.6021 |
| Thresholded | 0.2502 | 0.2527 | 0.6923 | 0.8000 | 0.8118 |
| *Baseline | <u>0.2502</u> | 0.2502 | 0.2502 | 0.2502 | 0.2502 |

**Table 5. Mean relative error for MNIST experiments.** The * symbol indicates methods we propose. Best results are underlined.

| Method | Target ranks $r$ | | | | |
|---|---|---|---|---|---|
| | 1 | 2 | 3 | 4 | 5 |
| *DA | 0.5796 | 0.3832 | 0.2673 | 0.1983 | 0.1522 |
| *DA+ALS | 0.5796 | 0.3832 | 0.2672 | 0.1982 | 0.1522 |
| Penalized | 0.6070 | 0.4072 | 0.2951 | 0.2238 | 0.1836 |
| Thresholded | 0.5872 | 0.4141 | 0.3171 | 0.2797 | 0.2738 |
| *Baseline | 0.8684 | 0.7484 | 0.6400 | 0.5476 | 0.4655 |

## Real data

In the first experiment on real data we consider the MNIST handwritten digits dataset [37] (available at http://yann.lecun.com/exdb/mnist/). We consider ten instances of each digit 0–9. The digits are 28 × 28 grayscale images with pixel values in the range [0, 255], where 0 represents white and 255 represents black. We make these binary by setting values less than 50 to 0, and values greater than or equal to 50 to 1. We apply BMF to the digits with target ranks $r \in$ [5]. The QUBO problem in DA BMF and DA+ALS BMF has $(28 + 28 + 28^2)r = 840r$ binary variables. Table 5 presents the mean relative error for each method across all instances of all digits. Fig 1 shows an example of the binary digit 3 and the low-rank approximations given by our DA+ALS BMF method.

In the second experiment on real data, we consider two gene expression datasets for two types of cancer: leukemia and malignant melanoma. The first dataset (available at https://www.pnas.org/content/101/12/4164) contains 38 gene samples for two kinds of leukemia, one of which can be further split into two subtypes [38]. The second dataset (available at https://schlieplab.org/Static/Supplements/CompCancer/CDNA/bittner-2000/) contains 38 gene samples, 31 of which are melanomas and 7 of which are controls [39]. We make these datasets binary by using the same thresholding approach as [3] which we describe here briefly. Let $0 \le c_1 < c_2$ be two real numbers. For a matrix $A \in \mathbb{R}^{m \times n}$ with nonnegative entries, let

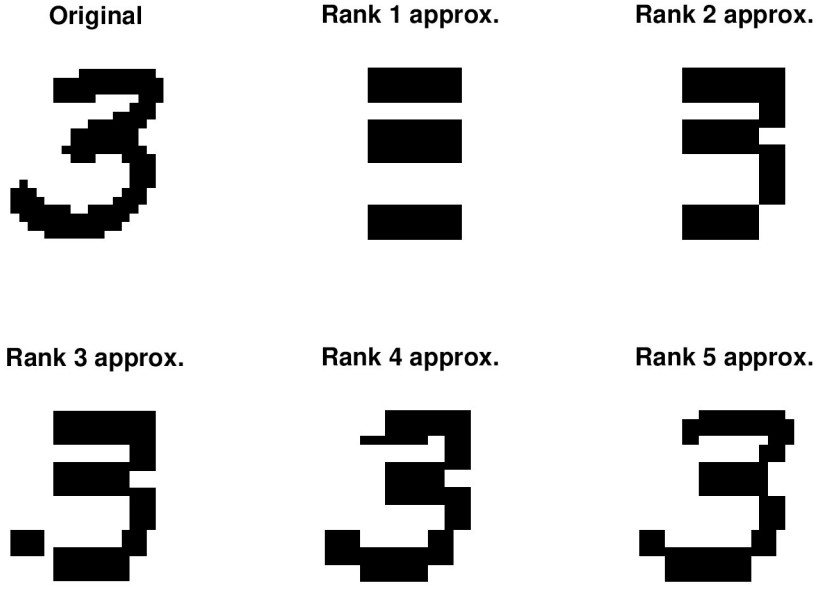

**Fig 1. Binary low rank approximation to MNIST digit using DA+ALS BMF.**

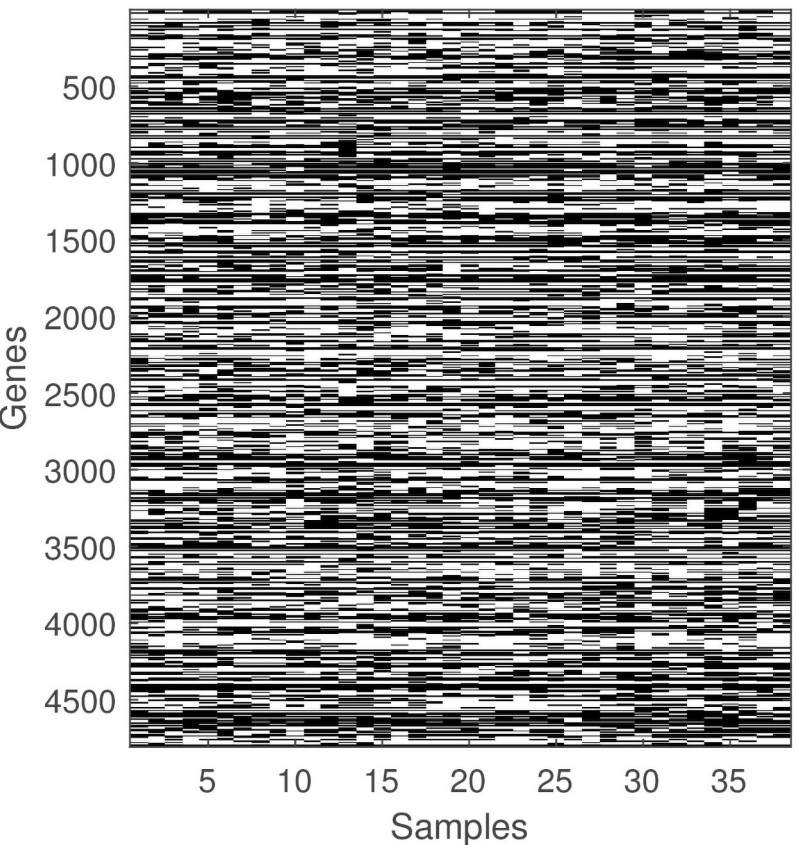

**Fig 2. The thresholded leukemia data.** Black entries are 1 and white entries are 0.

$\kappa \stackrel{\text{def}}{=} \sum_{ij} a_{ij}/(mn)$. $\tilde{A}$, a discretized version of $A$, is computed by setting its entries

$$\tilde{a}_{ij} = \begin{cases} 1 & \text{if } a_{ij} \leq \kappa c_1 \text{ or } a_{ij} \geq \kappa c_2, \\ 0 & \text{otherwise.} \end{cases}$$

Optionally, the columns of $A$ can be normalized so that they have unit Euclidean norm prior to discretization. As an additional step, we remove any rows from $\tilde{A}$ that contain only zeros. Following [2, 3], we use $c_1 = 1/7$ and $c_2 = 5$ without column normalization on the leukemia dataset. The melanoma dataset contains negative entries. We therefore first add a constant $-\min_{ij} a_{ij}$ to all entries in $A$. Then, we normalize the columns of the resulting matrix and discretize using $c_1 = 0.96$ and $c_2 = 1.04$. Fig 2 shows what the thresholded leukemia data looks like.

After thresholding and removing any rows that are all zero, the two datasets are matrices of size $4806 \times 38$ and $2201 \times 38$, respectively. We use the sampling technique in the section Handling large rectangular matrices for both datasets with a sample size of $s = 30$. Furthermore, we include the constraints on the $V$ matrix in the QUBO formulations as discussed in the section Useful constraints for data analysis. Although we expect that this will increase the decomposition error somewhat, including such constraints can be helpful e.g. when clustering the samples. The QUBO problem in our methods has $(30 + 38 + 30 \cdot 38)r = 1208r$ binary variables. Table 6 reports the mean relative error across 10 trials for each dataset.

**Table 6. Mean relative error for gene expression data.** The * symbol indicates methods we propose. Best results are <u>underlined</u>.

| Method | Target ranks $r$ (leukemia dataset [38]) | | | | |
|---|---|---|---|---|---|
| | 1 | 2 | 3 | 4 | 5 |
| *DA | 0.4292 | 0.4069 | 0.3853 | 0.4257 | 0.4025 |
| *DA+ALS | <u>0.3939</u> | <u>0.3806</u> | <u>0.3716</u> | <u>0.3683</u> | <u>0.3600</u> |
| Penalized | 0.3977 | 0.3952 | 0.4767 | 0.5399 | 0.5810 |
| Thresholded | <u>0.3939</u> | 0.4219 | 0.6195 | 0.6625 | 0.6894 |
| *Baseline | 0.9698 | 0.9408 | 0.9123 | 0.8842 | 0.8563 |
| **Method** | **Target ranks $r$ (melanoma dataset [39])** | | | | |
| | 1 | 2 | 3 | 4 | 5 |
| *DA | 0.8898 | 0.8392 | 0.7869 | 0.7850 | 0.7722 |
| *DA+ALS | <u>0.8572</u> | <u>0.7757</u> | <u>0.7416</u> | <u>0.7252</u> | <u>0.7113</u> |
| Penalized | 0.8662 | 0.8405 | 0.8110 | 0.7903 | 0.7633 |
| Thresholded | 0.8662 | 0.8338 | 0.8226 | 0.8120 | 0.8116 |
| *Baseline | 0.9504 | 0.9027 | 0.8569 | 0.8132 | 0.7695 |

## Discussion

The accuracy improvement of DA+ALS BMF over DA BMF is typically very small, but more substantial in a few cases. Our DA+ALS BMF has the same or better accuracy as either of the methods by [2, 3] in 72 of the 75 experiments reported in Tables 1–6 in this paper, and a strictly better accuracy in 57 of the experiments.

For a fixed rank, the number of binary variables for the QUBO problem in DA BMF is similar across all experiments. The anneal time, which is the time spent by the DA looking for a solution, is about 40 seconds for the largest problems. The additional ALS steps used for DA +ALS take on average much less than a second when $m = 30$, and less than about 3 seconds when $m = 2000$. For $m = 50000$, the additional time for ALS can be more substantial, adding on average as much as 66 seconds for rank 5 experiments.

Two advantages of the methods by [2, 3] are that they typically are quite fast and that they can run on a standard computer. For all experiments except the synthetic ones with $m = 50000$, their penalized and thresholded algorithms run in less than 2 and 20 seconds on average, respectively. The very large experiments with $m = 50000$ can take longer, up to 39 and 289 seconds on average for the penalized and thresholded algorithms, respectively, when $r = 5$.

Based on these observations, DA+ALS BMF seems like the superior method when accuracy is crucial. The methods by [2, 3] may be more suitable when speed and accessibility are more important. With that said, we believe that the DA could be run with many fewer iterations with little or no degradation in performance in most of our experiments. The trade-off between accuracy and speed for the DA, and how to choose the number of iterations to strike a good balance, are interesting directions for future research.

Certain matrices, like those of size 2000 × 30 and expected density 0.2 considered in Table 2, seem inherently difficult to handle for any of the sophisticated methods. Indeed, it is surprising that the simple baseline method substantially outperforms all other methods.

## Conclusion

BMF has many applications in data mining. We have presented two ways to formulate BMF as QUBO problems. These formulations can be used to do BMF on special purpose hardware, such as the D-Wave quantum annealer and the Fujitsu DA. We also discussed how clustering constraints can easily be incorporated into our QUBO formulations. Moreover, we showed

how sampling and alternating binary least squares can be used to handle large rectangular matrices. Our experiments, which we run on the Fujitsu DA, are encouraging and show that our proposed methods typically give more accurate solutions than competing methods.

The special purpose hardware technologies discussed in this paper are still in an early phase of development. As these technologies mature, we believe that they will emerge as powerful tools for solving problems in data mining and other areas.

## Supporting information

**S1 Text. Supplementary material.** Contains supporting text to the main manuscript. (ZIP)

## Author Contributions

**Conceptualization:** Osman Asif Malik, Hayato Ushijima-Mwesigwa, Arnab Roy, Avradip Mandal, Indradeep Ghosh.

**Data curation:** Osman Asif Malik.

**Formal analysis:** Osman Asif Malik.

**Investigation:** Osman Asif Malik, Hayato Ushijima-Mwesigwa, Arnab Roy, Avradip Mandal, Indradeep Ghosh.

**Methodology:** Osman Asif Malik, Hayato Ushijima-Mwesigwa, Arnab Roy, Avradip Mandal, Indradeep Ghosh.

**Project administration:** Indradeep Ghosh.

**Resources:** Hayato Ushijima-Mwesigwa.

**Software:** Osman Asif Malik, Hayato Ushijima-Mwesigwa.

**Supervision:** Indradeep Ghosh.

**Validation:** Osman Asif Malik, Hayato Ushijima-Mwesigwa.

**Visualization:** Osman Asif Malik.

**Writing – original draft:** Osman Asif Malik, Hayato Ushijima-Mwesigwa.

**Writing – review & editing:** Osman Asif Malik, Hayato Ushijima-Mwesigwa, Arnab Roy, Avradip Mandal, Indradeep Ghosh.

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
