## [Decision Letter · Decision Letter 0]

16 Jun 2021

PONE-D-21-17581

Binary matrix factorization on special purpose hardware

PLOS ONE

Dear Dr. Malik,

Thank you for submitting your manuscript to PLOS ONE. After careful consideration, we feel that it has merit but does not fully meet PLOS ONE’s publication criteria as it currently stands. Therefore, we invite you to submit a revised version of the manuscript that addresses the points raised during the review process.

Based on the comments received from the reviewers and my own observation, I recommend major revisions for the article.

We look forward to receiving your revised manuscript.

Kind regards,

Thippa Reddy Gadekallu

Academic Editor

PLOS ONE

Journal Requirements:

"I have read the journal's policy and the authors of this manuscript have the following competing interests: The authors HUM, AR, AM and IG are employees of Fujitsu Laboratories of America, Inc., which is the developer of the Fujitsu Digital Annealer used in the experiments. OAM worked as an intern at Fujitsu Laboratories of America, Inc., when the research in this paper was carried out."

We note that one or more of the authors are employed by a commercial company: Fujitsu Laboratories of America, Inc..

2.1. Please provide an amended Funding Statement declaring this commercial affiliation, as well as a statement regarding the Role of Funders in your study. If the funding organization did not play a role in the study design, data collection and analysis, decision to publish, or preparation of the manuscript and only provided financial support in the form of authors' salaries and/or research materials, please review your statements relating to the author contributions, and ensure you have specifically and accurately indicated the role(s) that these authors had in your study. You can update author roles in the Author Contributions section of the online submission form.

2.2. Please also provide an updated Competing Interests Statement declaring this commercial affiliation along with any other relevant declarations relating to employment, consultancy, patents, products in development, or marketed products, etc.  

Reviewers' comments:

Reviewer's Responses to Questions

**Comments to the Author**

1. Is the manuscript technically sound, and do the data support the conclusions?

Reviewer #1: Yes

Reviewer #2: Partly

2. Has the statistical analysis been performed appropriately and rigorously? 

Reviewer #1: Yes

Reviewer #2: Yes

3. Have the authors made all data underlying the findings in their manuscript fully available?

Reviewer #1: Yes

Reviewer #2: No

4. Is the manuscript presented in an intelligible fashion and written in standard English?

Reviewer #1: Yes

Reviewer #2: Yes

5. Review Comments to the Author

Reviewer #1: Authors propose a sampling based approach to overcome

this challenge, allowing us to factorize large rectangular matrices. The authors run experiments

on the Fujitsu Digital Annealer, a quantum inspired CMOS annealer, on both synthetic

and real data, including gene expression data. These experiments show that there approach is able to produce more accurate BMFs than competing methods.

The paper is well written and however there are some minor corrections which can corrected. The authors should define all the notations and acronyms before using them.

BLS problems can be solved via a QUBO formulation can be explained in two ot three lines.

Add the advantages of the proposed system in one quoted line for justifying the proposed approach in the Introduction section.

•The motivation for the present research would be clearer, by providing a more direct link between the importance of choosing your own method.

Reviewer #2: The paper proposes "Binary matrix factorization on special purpose hardware". The topic is interesting. However, the paper does not explicitly show novelty and originality. More in-depth discussion of the finding is required. Also, several sections should be improved in terms of quality representation.

1. Author failed to highlight contributions in this article.

2. How is the paper different from existing works? What gaps they identified in existing works? How the proposed approach solves the research/scientific problems identified in the existing works? Compare the results with recent state of the art.

3. Authors are used baseline method but there is no discussion about this abstract, introduction etc.

4. In discussion section authors stated that “Our DA+ALS BMF has the same or better accuracy as 333 either of the methods by [2, 3] in 72 of the 75 experiments reported in the tables in this paper, and a strictly better accuracy in 57 of the experiments”. But I can’t find any table that reported these experiments.

5. Improve introduction and related work sections for readability purpose.

6. Motivation and contributions must well written in introduction section.

6. PLOS authors have the option to publish the peer review history of their article (what does this mean?). If published, this will include your full peer review and any attached files.

Reviewer #1: No

Reviewer #2: **Yes: **Kadiyala Ramana

---

## [Author Response · Author response to Decision Letter 0]

14 Jul 2021

Please see the attached "Response to Reviewers.pdf" for our response to the academic editor and the reviewers

---

## [Decision Letter · Decision Letter 1]

26 Nov 2021

Binary matrix factorization on special purpose hardware

PONE-D-21-17581R1

Dear Dr. Malik,

We’re pleased to inform you that your manuscript has been judged scientifically suitable for publication and will be formally accepted for publication once it meets all outstanding technical requirements.

Kind regards,

Thippa Reddy Gadekallu

Academic Editor

PLOS ONE

Additional Editor Comments (optional):

Reviewers' comments:

Reviewer's Responses to Questions

**Comments to the Author**

1. If the authors have adequately addressed your comments raised in a previous round of review and you feel that this manuscript is now acceptable for publication, you may indicate that here to bypass the “Comments to the Author” section, enter your conflict of interest statement in the “Confidential to Editor” section, and submit your "Accept" recommendation.

Reviewer #2: All comments have been addressed

2. Is the manuscript technically sound, and do the data support the conclusions?

Reviewer #2: (No Response)

3. Has the statistical analysis been performed appropriately and rigorously? 

Reviewer #2: (No Response)

4. Have the authors made all data underlying the findings in their manuscript fully available?

Reviewer #2: (No Response)

5. Is the manuscript presented in an intelligible fashion and written in standard English?

Reviewer #2: (No Response)

6. Review Comments to the Author

Reviewer #2: (No Response)

7. PLOS authors have the option to publish the peer review history of their article (what does this mean?). If published, this will include your full peer review and any attached files.

Reviewer #2: **Yes: **Dr. Kadiyala Ramana